# Diagnostic Performance of Neutrophil to Lymphocyte Ratio, Monocyte to Lymphocyte Ratio, Platelet to Lymphocyte Ratio, and Platelet to Mean Platelet Volume Ratio in Periprosthetic Hip and Knee Infections: A Systematic Review and Meta-Analysis

**DOI:** 10.3390/diagnostics12092033

**Published:** 2022-08-23

**Authors:** Enrico Festa, Tiziana Ascione, Alessio Bernasconi, Donato Di Gennaro, Morena Anna Basso, Amedeo Guarino, Giovanni Balato

**Affiliations:** 1Orthopedic Unit, Department of Public Health, Federico II University Naples, 80131 Naples, Italy; 2Service of Infectious Disease, Department of Medicine, Cardarelli Hospital Naples, 80131 Naples, Italy

**Keywords:** neutrophils-to-lymphocytes ratio, platelets-to-lymphocytes ratio, monocytes-to-lymphocytes ratio, platelets-to-medium platelet volume ratio, diagnosis, periprosthetic joint infection, total knee arthroplasty

## Abstract

The current literature on the diagnosis of periprosthetic joint infection provides controversial evidence on the diagnostic accuracy of MLR, NLR, PVR, and PLR. Therefore, this critical literature search and meta-analysis was aimed to summarize the diagnostic accuracy of these biomarkers for the diagnosis of hip and knee prosthetic infection. According to the PRISMA flowchart, we searched MEDLINE, Scopus, and Web of Science, for studies on these ratios for diagnosing PJI. Sensitivity, specificity, positive and negative likelihood ratio, diagnostic odds ratio, and AUC were analyzed. We included 11 articles in our meta-analysis, including 7537 patients who underwent total hip and knee arthroplasties; among these, 1974 (26%) patients reported a joint infection. The pooled sensitivity and specificity were 0.72 and 0.74, respectively, for NLR, 0.72 and 0.77 for PVR, and 0.77 and 0.75 for PLR. The sensitivity of MLR ranges from 0.54 to 0.81, while the specificity ranges from 0.78 to 0.81. Regarding the evaluation of AUCs, the best diagnostic performance was achieved by MLR (AUC = 0.77) followed by PLR (AUC = 0.75), NLR (AUC = 0.73), and PVR (AUC = 0.70). This meta-analysis demonstrates a fair diagnostic accuracy of these ratios, thus not being useful as a screening tool.

## 1. Introduction

Periprosthetic joint infection (PJIs) is one of the most severe complications after prosthesis implantation. Despite different guidelines and diagnostic criteria proposed to detect a septic process accurately, PJI diagnosis remains a significant challenge for orthopedic surgeons [1]. Nowadays, to diagnose the presence of infection properly, we rely on the definition of prosthetic infection proposed by the second International Consensus Meeting on PJI [2,3]. simply obtained from the routine blood test, some of these are valuable in the blood, such as serum CRP [3,4,5], D-dimer [6,7,8], and erythrocyte sedimentation rate [3,4,5], others in synovial fluid like leukocyte esterase [9,10,11,12], synovial alpha defensin [13,14,15,16,17,18], elevated synovial fluid white blood cell count [5,14,19,20,21], and the polymorphonuclear percentage [5,19,20,21]. None of these criteria alone achieve a high diagnostic accuracy [2,3,4,5,6,7,8,9,10,11,12,13,14,15,16]. For this reason, the literature has focused on new cheap, fast, and minimally invasive biomarkers. Platelet counts and their ratio with blood cells have aroused considerable interest [22,23,24,25,26] in responding to foreign organisms [23,27,28]. Indeed, the mean platelet volume is reported to be altered in various pathologies related to inflammatory states [29,30]. Furthermore, the role of polymorphonuclear leukocytes in inflammation as mediators and active agents is also well known. The number of polymorphonuclear leukocytes is well associated with infectious processes. A reduction in their number is associated with an increased risk of infection and infection-related death [31]. It has recently been described that the states of systemic inflammation and infectious processes influence not only the number of leukocytes and platelets but also the ratios between them, such as monocyte-lymphocyte ratio (MLR), neutrophil-lymphocyte ratio (NLR), platelet-lymphocyte ratio (PLR), and platelet-mean platelet volume ratio (PVR) [32,33,34,35,36,37]. These ratios represent inexpensive, easily available, and less invasive parameters than synovial markers, requiring a simple blood count for their analysis. Several studies have assessed the diagnostic accuracy of these parameters in this clinical setting with discordant results. To our knowledge, no systematic literature review on these ratios has been made. Therefore, we conducted a metanalysis to further address the diagnostic role of NLR, MLR, PLR, and PVR in PJI.

## 2. Materials and Methods

### 2.1. Search Strategy and Criteria

This systematic review was conducted according to the guidelines of the Preferred Reporting Items for Systematic Review and Meta-Analyses (PRISMA) [38] up to June 2022. We searched for studies investigating diagnostic accuracy and the diagnostic role of PLR, PVR, MLR, and NLR in patients with PJI in electronic databases, namely MEDLINE, Scopus, and Web of Science. A combination of the following keywords was used for the search: (“NLR” OR “Neutrophil-to-lymphocyte ratio” OR “neutrophil-lymphocyte ratio”) AND (“MLR” OR “Monocyte-to-lymphocyte ratio” OR “Monocyte-lymphocyte ratio”) AND (“PLR” OR “Platelet-to-lymphocyte ratio” OR “platelet-lymphocyte ratio”) AND (“PVR” OR “Platelet-to-mean platelet volume ratio” OR “platelet-mean platelet volume ratio”) AND (“PJI” OR “periprosthetic infection” OR “periprosthetic joint infection”) AD (“diagnosis” OR “detection”). No language or date restrictions were applied to our inclusion criteria. The selected articles’ reference list was also hand-searched to identify additional studies that could not be found initially with the search criteria. Longitudinal studies (retrospective and prospective) and randomized controlled trials evaluating the diagnostic accuracy of these ratios in PJI was finally selected. The exclusion criteria included: case reports, expert opinions, previous systematic reviews, letters to the editor, studies that did not report quantitative values of sensitivity, specificity or likelihood ratios, or diagnostic accuracy, and studies considering different diagnostic criteria concerning MSIS and ICM as a reference standard to rule out.

### 2.2. Study Assessment and Data Extraction

Titles and abstracts of all retrieved documents were initially screened by two independent reviewers. The full text was obtained for all documents that appeared to meet the inclusion criteria or those with uncertainty. Each study was then assessed according to our inclusion criteria by two independent reviewers, and any discrepancies regarding the eligibility of an article were resolved with a third author. The following relevant data were extracted from each included study: patient demographics, sample size, type of arthroplasty, sensitivity, specificity, positive (PLR) and negative likelihood ratio (NLR), and diagnostic odds ratio (DOR). Two authors performed the quality assessment of each study using QUADAS (Quality Assessment of Diagnostic Accuracy Studies) [39]. The QUADAS score consists of four domains: (1) patient selection; (2) index test, reference standard; (3) flow; (4) timing. The risk of bias assessment of the four domains was assessed with signaling questions. Questions were answered, “yes” for low risk of bias/concerns, “no” for high risk of bias/concerns, or “unclear.”

### 2.3. Statistical Analysis

All analysis was conducted using the OpenMetaAnalyst software version 12.11.14 (Brown University, Providence, RI, USA) and SPSS version 23 (SPSS Inc., Chicago, IL, USA) for all statistical analyses. *p* ⊴ 0.05 was considered significant. The sensitivity, specificity, PLR and NLR, and DOR were meta-analyzed using the bivariate diagnostic random-effects model described by Reitsma et al. [40]. Heterogeneity between studies was assessed with I2 statistic (0–40%: not relevant; 30–60%: moderate; 50–90%: substantial; 75–100%: high) [41]. The area under the curve (AUC) was calculated using the trapezoidal rule, including the extrapolated points of the ROC curve. We adopted a meta-analytic approach if the number of studies was more elevated than five, given the greater power in the overall estimation of an effect. On the contrary, we proceeded not to describe the pooled results but only range if the number of studies was fewer than five.

## 3. Results

The flow chart diagram of the selection process is summarized in Figure 1. Computer search and manual screening of reference lists of relevant studies identified 154 potentially relevant citations. After an initial screening of titles and abstracts, the full text of 15 articles was evaluated, leading to the elimination of four further documents. The remaining 11 articles were included in our meta-analysis. These included articles were retrospective investigations [22,23,42,43,44,45,46,47,48,49,50]. Table 1 summarizes the characteristics of the studies. Overall, 7537 (50% female) patients undergoing TJA were evaluated, among whom 1974 (26%; range 16–50%) were confirmed to have a joint infection according to ICM 2013, MSIS, EBJS criteria, and ICM 2018. The site of PJI was reported in all studies involving patients affected by hip or knee prosthetic infections. Only three articles analyzed all four parameters [22,42,47] and all the included articles gave their cut-offs for discriminating PJI and aseptic failures for the analyzed ratios. The results of the QUADAS-2 assessment for each study are provided in Table 2.

### 3.1. Diagnostic Accuracy of MLR

This parameter has been analyzed in four of the eleven included articles [22,42,46,47]. One thousand seven hundred ninety-one patients with a percentage of PJI of 43% (767/1791) were evaluated. Zhao et al. have not been considered because they studied the lymphocyte-to-monocyte ratio (LMR) [49]. The sensitivity, specificity, positive LR and negative LR, and DOR of the studies are shown in Table 3. The sensitivity ranges from 0.54 to 0.81, while specificity ranges from 0.78 to 0.81. Furthermore, positive and negative likelihood ratios and DOR vary from 2.6 to 3.9, 0.23 to 0.58, and 4.45 to 13.19, respectively. In addition, the AUC for MLR was 0.77 (95% CI 0.76–0.79).

### 3.2. Diagnostic Accuracy of NLR

Neutrophil-to-lymphocyte ratio has been analyzed in all of the included studies [22,42,43,44,45,46,47,48,49,50] but one [23]. The number of patients included in these studies is 2600, with a percentage of PJI of 39% (1025/2600). The sensitivity, specificity, positive LR and negative LR, and DOR of the included studies, along with their corresponding pooled indices, are shown in Table 4. The pooled sensitivity and specificity were 0.72 (95% CI: 0.66–0.77) and 0.74 (95% CI: 0.70–0.79), respectively. The heterogeneity (I2 statistics) for sensitivity was 69.5 p<0.001 and for specificity was 74.21% p<0.001, thus reflecting substantial heterogeneity (Figure 2 and Figure 3). The pooled positive LR, negative LR, DOR were 2.8 (IC 95%: 2.30–3.54), 0.33 (IC 95%: 0.25–0.44), and 8.13 (IC 95%: 5.13–12.89), respectively. In addition, the AUC for NLR was 0.73 (95% CI 0.71–0.75).

### 3.3. Diagnostic Accuracy of PVR

Platelet-to-mean platelet volume ratio has been studied in five articles [22,23,42,45,47]. The number of patients included in these studies is 6363, with a percentage of PJI of 24% (1546/6363). The sensitivity, specificity, positive LR and negative LR, and DOR of the included studies, along with their corresponding pooled indices, are shown in Table 5. The pooled sensitivity and specificity were 0.72 (95% CI: 0.49–0.87) and 0.77 (95% CI: 0.73–0.81), respectively. The heterogeneity (I2 statistics) for sensitivity was 97.75 p<0.001 and for specificity was 74.76% p<0.003, thus reflecting substantial heterogeneity (Figure 4 and Figure 5). The pooled positive LR, negative LR, DOR were 2.9 (IC 95%: 2.41–3.58), 0.33 (IC 95%: 0.16–0.66), and 8.73 (IC 95%: 3.71–20.54), respectively. In addition, the AUC for PVR was 0.70 (95% CI 0.69–0.72).

### 3.4. Diagnostic Accuracy of PLR

Platelet-to-lymphocyte ratio has been studied in five articles [22,42,46,47,50]. The number of patients included in these studies is equal to 1895, with a percentage of PJI of 42% (793/1895). The sensitivity, specificity, positive LR and negative LR, and DOR of the included studies, along with their corresponding pooled indices, are shown in Table 6. The pooled sensitivity and specificity were 0.77 (95% CI: 0.73–0.82) and 0.75 (95% CI: 0.60–0.85), respectively. The heterogeneity (I2 statistics) for sensitivity was 48.89 p<0.098 and for specificity was 95.34% p<0.001, thus reflecting substantial heterogeneity (Figure 6 and Figure 7). The pooled positive LR, negative LR, DOR were 3.3 (IC 95%: 1.2–5.35), 0.3 (IC 95%: 0.20–0.40), and 11.16 (IC 95%: 5–24.9), respectively. In addition, the AUC for PLR was 0.75 (95% CI 0.73–0.77).

## 4. Discussion

Recently, different studies investigated serum biomarkers such as NLR, PLR, PVR, and MLR with discordant results regarding diagnostic accuracy. Our meta-analysis tried to clarify these biomarkers’ role in the complex diagnostic workup in hip and knee prosthetic infections. The pooled estimation of the 11 studies included in our study indicates a similar result in terms of sensitivity and specificity for the following ratios: NLR, PVR, and PLR. For MLR, we did not evaluate the pooled sensitivity, specificity, positive LR, negative LR, and DOR because the studies included were less than five [22,42,46,47]. The false negative result rates for MLR, NLR, PVR and PLR were 32% (248/767), 29% (301/1025), 39% (613/1546) and 23% (183/793), respectively. The pooled sensitivity was 0.72 for NLR, 0.72 for PVR, and 0.77 for PLR. The sensitivity of MLR ranges from 0.54 to 0.81. However, it is still unclear why some patients with joint infection had normal ratio values at presentation, as evidenced in some of the studies included in our pooled analysis. Low-grade infection and biofilm formation characterized by reduced immunologic and inflammatory responses could partially explain the false negative results [51]. The false positive result rates were 20% (210/1024) for MLR, 25% (398/1575) for NLR, 20% (966/4817) for PVR and 26% (295/1102) for PLR. Considering these results, the pooled specificity was 0.74, 0.77, and 0.75 for NLR, PVR, and PLR, respectively. Different cutoffs could explain this metanalysis’s different specificities for PLR and PVR. It is well known that lowering a test’s cutoff decreases its specificity. Indeed, Maimati Z. et al. [42] and Xu H. et al. [46] reported similar PLR thresholds and consequently a comparable specificity of 0.57 and 0.57, respectively. In contrast, Tirumala V. et al. [22] and Klemt C. et al. [47] reported a higher cutoff value for PLR, thus describing a specificity of 0.82 and 0.83, respectively. Furthermore, a low number of patients affected by acute periprosthetic joint infections described by Zhao G. et al. [49] could explain the high specificity (0.87), despite the proposed cutoff value being 139.22. Similarly, Paziuk T. et al. [23] and Sigmund IK. et al. [45] reported the optimal threshold for PVR of 31.7 and 29.4 with a specificity of 80.85 and 81.2, respectively. On the contrary, Maimati et al. [42] reported a lower cutoff value with lower specificity. The specificity of MLR ranges from 0.78 to 0.81. Many studies reported that MLR, NLR, PVR, and PLR are reasonably higher in conditions such as bacteremia or other infections [26,52,53]. The reason for the increase in MLR, NLR, and PLR is that during the process of sepsis or any disease, the various anti-inflammatory cytokines released into the bloodstream can induce immunosuppression, subsequently leading to apoptosis of a large number of lymphocytes reducing the number of lymphocytes throughout the body [54,55]. The neutrophil count increases dramatically, and these neutrophils then migrate to the affected area; however, the lymphocyte count decreases due to immunosuppression. Neutrophils are also suggested to negatively influence T-cell activation, resulting in reduced lymphocyte-mediated immune response [56,57]. Furthermore, macrophage-derived chemokines induce macrophages precursor cells to polarize and migrate to inflammatory tissues to participate in the pathogenesis of inflammation [58]. Moreover, megakaryocytes accelerate their proliferation rate during systemic inflammation, resulting in thrombocytosis. It has long been known that platelets play a significant role in our body’s innate immune response [59]. This could explain the possible increase in PVR. All the evaluated parameters reported a positive likelihood of 2 to 5 and a negative likelihood of 0.3. They reflect that a positive or negative result for these criteria indicates a small increased or decreased probability of PJI. On the contrary, the DOR > 1, emerging from our meta-analysis for all the serum biomarkers, is indicative of better test performance. Regarding the evaluation of AUCs, the best diagnostic performance was achieved by MLR (AUC = 0.77), followed by PLR (AUC = 0.75), NLR (AUC = 0.73), and PVR (AUC = 0.70). As specifically concerns sensitivity, univariate meta-regression has evidenced two main sources of heterogeneity, i.e., sample size and reference standard for diagnosis of PJI. The heterogeneity of the former aspect (i.e., sample size) is largely attributable to a wide range of patients included in the different studies, with the article of Paziuk et al. accounting for the larger extent [23]. The heterogeneity of the reference standard originates instead from the different definitions of PJI. The lack of a threshold effect is another important aspect that emerged in our meta-analysis. There are any cut-offs available yet, all the studies included proposed their own cut-off for the ratios analyzed and this aspect increased the heterogeneity. The strengths and potential limitations of this study should be acknowledged. This study is the first metanalysis on the utility of MLR, PVR, PLR, and NLR in hip and knee prosthetic infections. We adopted stringent eligibility criteria that led to the exclusion of studies that assessed MLR, PVR, PLR, and NLR results in patients with a prosthetic infection that differed from TKA and THA. This study has a few drawbacks. First, this meta-analysis was performed in retrospective studies because of the lack of randomized controlled trials on the diagnostic accuracy of these biomarkers in patients suspected of prosthetic infection. Moreover, the different diagnostic criteria used to rule out PJI, the small number of patients included in the studies, and different cutoff values proposed for each parameter may have contributed to the heterogeneity among studies that emerged for some outcomes assessed in the present meta-analysis. The disparity of diagnostic criteria could alter the ability of these diagnostic tools to distinguish between septic and aseptic loosening and, consequently, change their accuracy. Furthermore, the lack of a threshold effect is another crucial aspect of our meta-analysis. All the studies included proposed their cutoff for the ratios analyzed, which increased the heterogeneity.

## 5. Conclusions

The available evidence suggests that MLR, NLR, PVR, and PLR reported a high rate of false positive and negative results. Indeed, this meta-analysis demonstrates a fair diagnostic accuracy of these ratios, thus not being useful as a screening tool. Further studies would be needed to validate their potential diagnostic role.

## Figures and Tables

**Figure 1 diagnostics-12-02033-f001:**
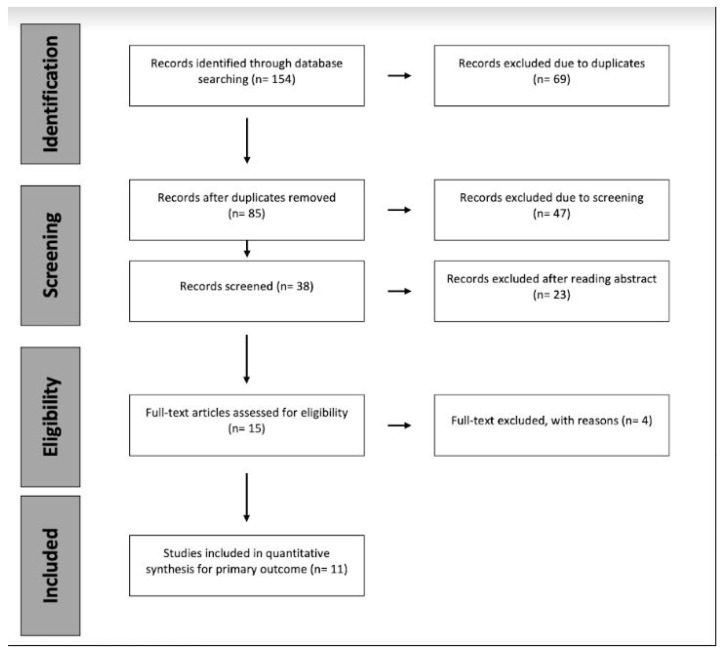
PRISMA flow chart diagram.

**Figure 2 diagnostics-12-02033-f002:**
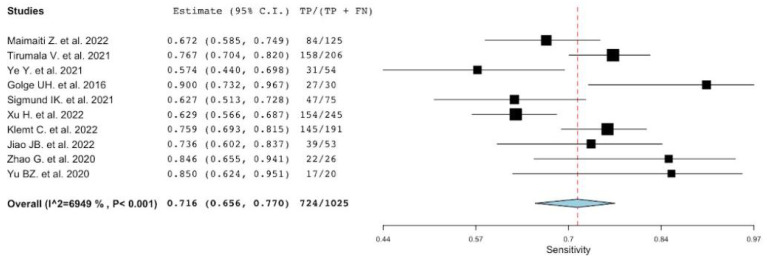
A plot showing the study-specific and meta-analyzed estimates for sensitivity for NLR [22,42,43,44,45,46,47,48,49,50].

**Figure 3 diagnostics-12-02033-f003:**
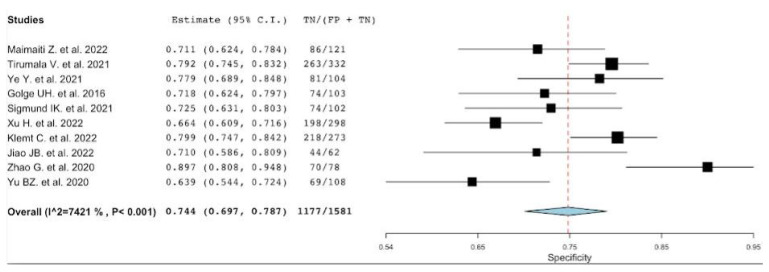
A plot showing the study-specific and meta-analyzed estimates for specificity for NLR [22,42,43,44,45,46,47,48,49,50].

**Figure 4 diagnostics-12-02033-f004:**
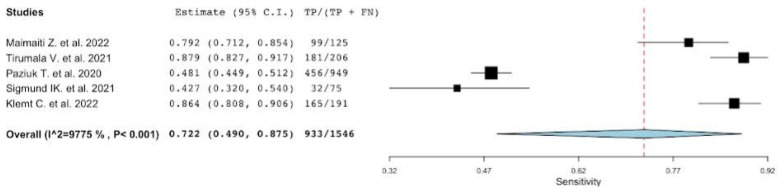
A plot showing the study-specific and meta-analyzed estimates for sensitivity for PVR [22,23,42,45,47].

**Figure 5 diagnostics-12-02033-f005:**
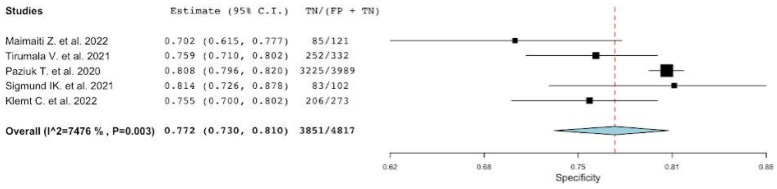
A plot showing the study-specific and meta-analyzed estimates for specificity for PVR [22,23,42,45,47].

**Figure 6 diagnostics-12-02033-f006:**
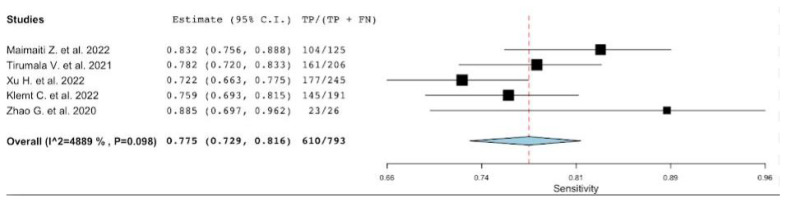
A plot showing the study-specific and meta-analyzed estimates for sensitivity for PLR [22,42,46,47,49].

**Figure 7 diagnostics-12-02033-f007:**
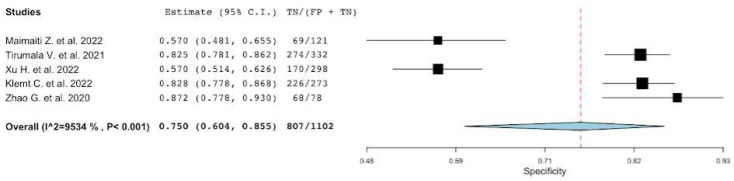
A plot showing the study-specific and meta-analyzed estimates for specificity for PLR [22,42,46,47,49].

**Table 1 diagnostics-12-02033-t001:** Characteristics of included studies.

Lead Author, Publication Date	Site of Arthroplasty in Infected Group Hip/Knee	No Patients ^a^ (PJIs n%)	Reference Standard	Parameters Evaluated MLR/NLR/MLR/PVR
Maimaiti Z. et al., 2022 [42]	57/68	125/246 (50%)	MSIS	MLR,NLR,PVR,PLR
Tirumala V. et al., 2021 [22]	0/206	206/538 (38%)	ICM 2018	MLR,NLR,PVR,PLR
Paziuk T. et al., 2020 [23]	NA	949/4938 (19%)	MSIS	PVR
Ye Y. et al 2021 [43]	27/27	54/158 (34%)	ICM 2018	NLR
Golge UH. et al., 2016 [44]	30/0	30/133 (22%)	NA	NLR
Sigmund IK. et al., 2021 [45]	36/39	75/177 (42%)	EBJS criteria	NLR,PVR
Xu H. et al., 2022 [46]	NA	245/543 (45%)	ICM 2013	MLR,NLR,PLR
Klemt C. et al., 2022 [47]	464/0	191/464 (41%)	ICM 2018	MLR,NLR,PVR,PLR
Jiao JB. et al., 2022 [48]	NA	53/115 (46%)	MSIS	NLR
Zhao G. et al., 2020 [49]	NA	26/104 (25%)	MSIS	NLR,PLR
Yu BZ. et al., 2020 [50]	37/17	20/121 (16%)	MSIS	NLR

^a^ The values were given as the number of PJI/total of joints involved in the study. NA = not available; MSIS = Muscoskeletal Infection Society; ICM = International Consensus Meeting; EBJS = European Bone and Joint Infection Society; MLR = Monocyte-Lymphocyte Ratio; NLR = Neutrophil-Lymphocyte Ratio, PVR = Platelet-Mean Platelet Volume ratio, PLR = Platelet-Lymphocyte Ratio.

**Table 2 diagnostics-12-02033-t002:** QUADAS-2 scores for studies included in the meta-analysis.

^a^ QUADAS-2 Score	1 *	2 *	3 *	Bias	Appl.	4 †	5 †	Bias	Appl.	6 ‡	7 ‡	Bias	Appl.	8 **	9 **	10 **	11 **	Bias
Maimaiti Z. et al., 2022 [42]	Yes	Yes	Yes	Low	Low	No	No	Low	Low	Yes	Yes	Low	Low	Yes	Yes	Yes	Yes	Low
Tirumala V. et al., 2021 [22]	Yes	Yes	Yes	Low	Low	No	No	Low	Low	Yes	Yes	Low	Low	Yes	Yes	Yes	Yes	Low
Paziuk T. et al., 2020 [23]	Yes	Yes	Yes	Low	Low	No	No	Low	Low	Yes	Yes	Low	Low	Yes	Yes	Yes	Yes	Low
Ye Y. et al. 2021 [43]	Yes	Yes	Yes	Low	Low	No	No	Low	Low	Yes	Yes	Low	Low	Yes	Yes	Yes	Yes	Low
Golge UH. et al., 2016 [44]	Yes	Yes	NC	NC	NC	No	No	Low	Low	Yes	Yes	Low	Low	Yes	Yes	Yes	Yes	Low
Sigmund IK. et al., 2021 [45]	Yes	Yes	Yes	Low	Low	No	No	Low	Low	Yes	Yes	Low	Low	Yes	Yes	Yes	Yes	Low
Xu H. et al., 2022 [46]	Yes	Yes	Yes	Low	Low	No	No	Low	Low	Yes	Yes	Low	Low	Yes	Yes	Yes	No	Low
Klemt C. et al., 2022 [47]	Yes	Yes	Yes	Low	Low	No	No	Low	Low	Yes	Yes	Low	Low	Yes	Yes	Yes	Yes	Low
Jiao JB. et al., 2022 [48]	Yes	Yes	Yes	Low	Low	No	No	Low	Low	Yes	Yes	Low	Low	Yes	Yes	Yes	No	Low
Zhao G. et al., 2020 [49]	Yes	Yes	Yes	Low	Low	No	No	Low	Low	Yes	Yes	Low	Low	Yes	Yes	Yes	Yes	Low
Yu BZ. et al., 2020 [50]	Yes	Yes	Yes	Low	Low	No	No	Low	Low	Yes	Yes	Low	Low	Yes	Yes	Yes	Yes	Low

^a^ QUADAS, a quality assessment tool for diagnostic accuracy studies; Bias, risk of bias; Appl, concerns regarding applicability NC, not clear. * Domain 1: Patient selection. Numbers correspond with the following questions: 1. Was a consecutive or random sample of patients enrolled? 2. Was a case-control design avoided? 3. Did the study avoid inappropriate exclusions? † Domain 2: Index test. Numbers correspond with the following questions: 4. Were the index test results interpreted without knowledge of the results of the reference standard? 5. If a threshold was used, was it pre-specified? ‡ Domain 3: Reference test. Numbers correspond with the following questions: 6. Is the reference standard likely to classify the target condition correctly? 7. Were the reference standard results interpreted without knowledge of the results of the index test? ** Domain 4: Flow and timing. Numbers correspond with the following questions: 8. Was there an appropriate interval between index test(s) and reference standard? 9. Did all patients receive a reference standard? 10. Did patients receive the same reference standard? 11. Were all patients included in the analysis?

**Table 3 diagnostics-12-02033-t003:** Characteristics of diagnostic studies for MLR with 95% confidence intervals.

	Sensitivity	Specificity	Positive LR	Negative LR	DOR
Maimaiti Z. et al., 2022 [42]	0.6 [0.51–0.68]	0.81 [0.73–0.87]	3.1 [2.22–4.68]	0.49 [0.33–0.73]	6.39 [3.59–11.39]
Tirumala V. et al., 2021 [22]	0.81 [0.76–0.86]	0.78 [0.74–0.82]	3.7 [3.03–4.66]	0.23 [0.19–0.29]	13.19 [8.70–19.9]
Xu H. et al., 2022 [46]	0.54 [0.47–0.6]	0.79 [0.74–0.83]	2.6 [2.02–3.32]	0.58 [0.45–0.75]	4.45 [3.05–6.48]
Klemt C. et al., 2022 [47]	0.75 [0.69–0.81]	0.80 [0.75–0.85]	3.9 [3.01–5.01]	0.31 [0.24–0.39]	12.72 [8.15–19.85]
Pooled	NA	NA	NA	NA	NA

**Table 4 diagnostics-12-02033-t004:** Characteristics of diagnostic studies for NLR with 95% confidence intervals.

	Sensitivity	Specificity	Positive LR	Negative LR	DOR
Maimaiti Z. et al., 2022 [42]	0.67 [0.58–0.75]	0.71 [0.62–0.78]	2.3 [1.71–3.15]	0.46 [0.34–0.63]	5.03 [2.93–8.66]
Tirumala V. et al., 2021 [22]	0.76 [ 0.70–0.82]	0.79 [0.74–0.83]	3.7 [2.95–4.61]	0.29 [0.23–0.37]	12.55 [8.26–19.05]
Ye Y. et al. 2021 [43]	0.57 [0.44–0.70]	0.77 [0.69–0.85]	2.6 [1.69–3.98]	0.55 [0.36–0.84]	4.75 [2.33–9.66]
Golge UH. et al., 2016 [44]	0.90 [0.73–0.97]	0.72 [0.62–0.60]	3.2 [2.30–4.45]	0.13 [0.1–0.19]	22.97 [6.46–81.59]
Sigmund IK. et al., 2021 [45]	0.62 [0.51–0.73]	0.72 [0.63–0.80]	2.3 [1.59–3.27]	0.51 [0.36–0.74]	4.44 [2.34–8.40]
Xu H. et al., 2022 [46]	0.62 [0.56–0.69]	0.66 [0.61–0.72]	1.9 [1.55–2.26]	0.56 [0.46–0.67]	3.35 [2.35–4.77]
Klemt C. et al., 2022 [47]	0.75 [0.69–0.81]	0.79 [0.75–0.84]	3.7 [2.93–4.83]	0.30 [0.23–0.39]	12.49 [8.01–19.49]
Jiao JB. et al., 2022 [48]	0.73 [0.60–0.84]	0.71 [0.59–0.81]	2.5 [1.66–3.86]	0.36 [0.24–0.57]	6.81 [2.99–15.47]
Zhao G. et al., 2020 [49]	0.84 [0.65–0.94]	0.90 [0.81–0.95]	8.2 [4.19–16.23]	0.17 [0.09–0.34]	48.12 [13.22–175.23]
Yu BZ. et al., 2020 [50]	0.85 [0.62–0.95]	0.68 [0.58–0.76]	2.7 [1.88–3.67]	0.22 [0.16–0.31]	11.85 [3.24–43.28]
Pooled	0.72 [0.66–0.77]	0.74 [0.70–0,79]	2.8 [2.30–3.54]	0.33 [0.25–0.44]	8.13 [5.13–12.89]

**Table 5 diagnostics-12-02033-t005:** Characteristics of diagnostic studies for PVR with 95% confidence intervals.

	Sensitivity	Specificity	Positive LR	Negative LR	DOR
Maimaiti Z. et al., 2022 [42]	0.79 [0.71–0.85]	0.68 [0.61–0.78]	2.4 [1.99–3.55]	0.3 [0.22–0.39]	8.99 [5.02–16.09]
Tirumala V. et al., 2021 [22]	0.88 [0.83–0.92]	0.76 [0.71–0.80]	3.6 [2.99–4.44]	0.16 [0.13–0.19]	22.81 [14–37.15]
Paziuk T. et al., 2020 [23]	0.48 [0.45–0.51]	0.81 [0.80–0.82]	2.5 [2.29–2.75]	0.64 [0.58–0.70]	3.90 [3.36–4.53]
Sigmund IK. et al., 2021 [45]	0.43 [0.32–0.54]	0.81 [0.73–0.88]	2.3 [1.41–3.71]	0.71 [0.43–1.14]	3.25 [1.65–6.39]
Klemt C. et al., 2022 [47]	0.86 [0.81–0.91]	0.75 [0.7–0.80]	3.5 [2.84–4.37]	0.18 [0.14–0.22]	19.51 [11.87–32.07]
Pooled	0.72 [0.49–0.87]	0.77 [0.73–0.81]	2.9 [2.41–3.58]	0.33 [0.16–0.66]	8.73 [3.71–20.54]

**Table 6 diagnostics-12-02033-t006:** Characteristics of diagnostic studies for PLR with 95% confidence intervals.

	Sensitivity	Specificity	Positive LR	Negative LR	DOR
Maimaiti Z. et al., 2022 [42]	0.83 [0.76–0.89]	0.57 [0.48–0.66]	1.9 [1.55–2.41]	0.29 [0.24–0.37]	6.57 [3.64–11.87]
Tirumala V. et al., 2021 [22]	0.78 [0.72–0.83]	0.82 [0.78–0.86]	4.5 [3.5–5.71]	0.26 [0.21–0.34]]	16.90 [10.94–26.12]
Xu H. et al., 2022 [46]	0.72 [0.66–0.77]	0.57 [0.51–0.63]	1.7 [1.44–1.96]	0.49 [0.42–0.57]	3.46 [2.41–4.96]
Klemt C. et al., 2022 [47]	0.76 [0.69–0.82]	0.83 [0.78–0.87]	4.4 [3.36–5.79]	0.29 [0.22–0.38]	15.16 [9.60–23.94]
Zhao G. et al., 2020 [49]	0.88 [0.70–0.96]	0.87 [0.78–0.93]	6.9 [3.81–12.51]	0.13 [0.07–0.24]	52.13 [13.19–20.6]
Pooled	0.77 [0.73–0.82]	0.75 [0.60–0.85]	3.3 [1.2–5.35]	0.3 [0.20–0.40]	11.16 [5–24.9]

## Data Availability

Datasets collected or analyzed during the current study are available from the corresponding author on request.

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
