# Peer review of "Diagnostic Performance of Neutrophil to Lymphocyte Ratio, Monocyte to Lymphocyte Ratio, Platelet to Lymphocyte Ratio, and Platelet to Mean Platelet Volume Ratio in Periprosthetic Hip and Knee Infections: A Systematic Review and Meta-Analysis"

_diagnostics, 2022, doi:10.3390/diagnostics12092033_

Round 1
Reviewer 1 Report
General Remarks
The study is well focused and well strucdtured.
The quality of written English needs mild improvement.
The search strategy and the statistical analysis are very well executed
Introduction
is well structured and focused on the main object of the study.
Line 24....valuable in the blood...............please rewrite
line 36......in literature..............................delete
The Results section is clear and the tables add to the analysis.
Line 161..........please provide reference
The first paragraph of the discussion should focus on the main results of the study.
Reviewer 2 Report
This paper systematically reviewed the literature of MLR, NLR, PVR and PLR in the diagnosis of hip and knee prosthesis infection, and analyzed the accuracy of these test indicators in the diagnosis of hip and knee prosthesis infection. It has good innovation, and the overall writing logic is clear and the wording is appropriate. But,the specificity of PLR and the specificity of PVR in different studies have great differences. It is recommended to further discuss the possible causes and their influence on the results.
Reviewer 3 Report
I like this paper a lot and think the surgeon community will accept the conclusions. We are all looking for a silver bullet, and that just isn't the normal biological system. While I think the science of using these methods is sound, it really isn't the level that we think we need. I think these ratios do have some value but just don't give the level of overall consistency for the final answer to displace other methods. I do not have the expertise to ultimately value the methods they assess, but the information is valuable to me a practicing surgeon.
Author Response
Dear Editor,
Thanks to the Referees for their careful review of our paper “Diagnostic performance of Neutrophil to Lymphocyte ratio, Monocyte to Lymphocyte ratio, Platelet to Lymphocyte ratio and Platelet to Mean Platelet Volume ratio in periprosthetic hip and knee infections: a systematic review and meta-analysis”. We modified the manuscript in response to the comments. You’ll find our point-by-point response below each of the reviewers. Comments and changes to the text in the manuscript have been evidenced as " Track changes" in Latex. We hope that the changes made meet satisfactorily the suggestions of Reviewers and look forward to hearing from you.
Yours sincerely
Dr. Enrico Festa